# Discovery of Natural Language Concepts in Individual Units of CNNs

**Seil Na[1], Yo Joong Choe[2], Dong-Hyun Lee[3], Gunhee Kim[1]**
[1]Seoul National University, [2]Kakao, [3]Kakao Brain
seil.na@vision.snu.ac.kr, yj.c@kakaocorp.com,
benjamin.lee@kakaobrain.com, gunhee@snu.ac.kr
https://github.com/seilna/CNN-Units-in-NLP

## Abstract

Although deep convolutional networks have achieved improved performance in many natural language tasks, they have been treated as black boxes because they are difficult to interpret. Especially, little is known about how they represent language in their intermediate layers. In an attempt to understand the representations of deep convolutional networks trained on language tasks, we show that individual units are selectively responsive to specific morphemes, words, and phrases, rather than responding to arbitrary and uninterpretable patterns. In order to quantitatively analyze such an intriguing phenomenon, we propose a concept alignment method based on how units respond to the replicated text. We conduct analyses with different architectures on multiple datasets for classification and translation tasks and provide new insights into how deep models understand natural language.

## 1    Introduction

Understanding and interpreting how deep neural networks process natural language is a crucial and challenging problem. While deep neural networks have achieved state-of-the-art performances in neural machine translation (NMT) (Sutskever et al., 2014; Cho et al., 2014; Kalchbrenner et al., 2016; Vaswani et al., 2017), sentiment classification tasks (Zhang et al., 2015; Conneau et al., 2017) and many more, the sequence of non-linear transformations makes it difficult for users to make sense of any part of the whole model. Because of their lack of interpretability, deep models are often regarded as hard to debug and unreliable for deployment, not to mention that they also prevent the user from learning about how to make better decisions based on the model's outputs.

An important research direction toward interpretable deep networks is to understand what their hidden representations learn and how they encode informative factors when solving the target task. Some studies including Bau et al. (2017); Fong & Vedaldi (2018); Olah et al. (2017; 2018) have researched on what information is captured by individual or multiple units in visual representations learned for image recognition tasks. These studies showed that some of the individual units are selectively responsive to specific visual concepts, as opposed to getting activated in an uninterpretable manner. By analyzing individual units of deep networks, not only were they able to obtain more fine-grained insights about the representations than analyzing representations as a whole, but they were also able to find meaningful connections to various problems such as generalization of network (Morcos et al., 2018), generating explanations for the decision of the model (Zhou et al., 2018a; Olah et al., 2018; Zhou et al., 2018b) and controlling the output of generative model (Bau et al., 2019).

Since these studies of unit-level representations have mainly been conducted on models learned for computer vision-oriented tasks, little is known about the representation of models learned from natural language processing (NLP) tasks. Several studies that have previously analyzed individual units of natural language representations assumed that they align a predefined set of specific concepts, such as sentiment present in the text (Radford et al., 2017), text lengths, quotes and brackets (Karpathy et al., 2015). They discovered the emergence of certain units that selectively activate to those specific concepts. Building upon these lines of research, we consider the following question: *What natural language concepts are captured by each unit in the representations learned from NLP tasks?*

Unit 108: legal, law, legislative

- Better legal protection for accident victims.
- These rights are guaranteed under law.
- This should be guaranteed by law.
- This legislative proposal is unusual.
- Animal feed must be safe for animal health.

Unit 711: should, would, not, can

- That would not be democratic.
- That would be cheap and it would not be right.
- This is not how it should be in a democracy.
- I hope that you would not want that!
- Europe can not and must not tolerate this.

Figure 1: We discover the most activated sentences and aligned concepts to the units in hidden representations of deep convolutional networks. Aligned concepts appear frequently in most activated sentences, implying that those units respond selectively to specific natural language concepts.

To answer this question, we newly propose a simple but highly effective concept alignment method that can discover which natural language concepts are aligned to each unit in the representation. Here we use the term *unit* to refer to each channel in convolutional representation, and *natural language concepts* to refer to the grammatical units of natural language that preserve meanings; *i.e.* morphemes, words, and phrases. Our approach first identifies the most activated sentences per unit and breaks those sentences into these natural language concepts. It then aligns specific concepts to each unit by measuring activation value of replicated text that indicates how much each concept contributes to the unit activation. This method also allows us to systematically analyze the concepts carried by units in diverse settings, including depth of layers, the form of supervision, and data-specific or task-specific dependencies.

The contributions of this work can be summarized as follows:

- We show that the units of deep CNNs learned in NLP tasks could act as a natural language concept detector. Without any additional labeled data or re-training process, we can discover, for each unit of the CNN, natural language concepts including morphemes, words and phrases that are present in the training data.

- We systematically analyze what information is captured by units in representation across multiple settings by varying network architectures, tasks, and datasets. We use VD-CNN (Conneau et al., 2017) for sentiment and topic classification tasks on Yelp Reviews, AG News (Zhang et al., 2015), and DBpedia ontology dataset (Lehmann et al., 2015) and ByteNet (Kalchbrenner et al., 2016) for translation tasks on Europarl (Koehn, 2005) and News Commentary (Tiedemann, 2012) datasets.

- We also analyze how aligned natural language concepts evolve as they get represented in deeper layers. As part of our analysis, we show that our interpretation of learned representations could be utilized at designing network architectures with fewer parameters but with comparable performance to baseline models.

## 2 RELATED WORK

### 2.1 INTERPRETATION OF INDIVIDUAL UNITS IN DEEP MODELS

Recent works on interpreting hidden representations at unit-level were mostly motivated by their counterparts in computer vision. In the computer vision community, Zhou et al. (2015) retrieved image samples with the highest unit activation, for each of units in a CNN trained on image recognition tasks. They used these retrieved samples to show that visual concepts like color, texture and object parts are aligned to specific units, and the concepts were aligned to units by human annotators. Bau et al. (2017) introduced BRODEN dataset, which consists of pixel-level segmentation labels for diverse visual concepts and then analyzed the correlation between activation of each unit and such visual concepts. In their work, although aligning concepts which absent from BRODEN dataset requires additional labeled images or human annotation, they quantitatively showed that some individual units respond to specific visual concepts.

On the other hand, Erhan et al. (2009); Olah et al. (2017); Simonyan et al. (2013) discovered visual concepts aligned to each unit by optimizing a random initial image to maximize the unit activation by gradient descent. In these cases, the resulting interpretation of each unit is in the form of optimized images, and not in the natural language form as the aforementioned ones. However, these continuous interpretation results make it hard for further quantitative analyses of discrete properties of representations, such as quantifying characteristics of representations with layer depth (Bau et al.,

2017) and correlations between the interpretability of a unit and regularization (Zhou et al., 2018a). Nevertheless, these methods have the advantage that the results are not constrained to a predefined set of concepts, giving flexibility as to which concepts are captured by each unit.

In the NLP domain, studies including Karpathy et al. (2015); Tang et al. (2017); Qian et al. (2016); Shi et al. (2016a) analyzed the internal mechanisms of deep models used for NLP and found intriguing properties that appear in units of hidden representations. Among those studies, the closest one to ours is Radford et al. (2017), who defined a unit as each element in the representation of an LSTM learned for language modeling and found that the concept of sentiment was aligned to a particular unit. Compared with these previous studies, we focus on discovering a much wider variety of *natural language concepts*, including any morphemes, words, and phrases all found in the training data. To the best our knowledge, this is the first attempt to discover concepts among all that exist in the form of natural language from the training corpus. By extending the scope of detected concepts to meaningful building blocks of natural language, we provide insights into how various linguistic features are encoded by the hidden units of deep representations.

## 2.2 ANALYSIS OF DEEP REPRESENTATIONS LEARNED FOR NLP TASKS

Most previous work that analyzes the learned representation of NLP tasks focused on constructing downstream tasks that predict concepts of interest. A common approach is to measure the performance of a classification model that predicts the concept of interest to see whether those concepts are encoded in representation of a input sentence. For example, Conneau et al. (2018); Adi et al. (2017); Zhu et al. (2018) proposed several probing tasks to test whether the (non-)linear regression model can predict well the syntactic or semantic information from the representation learned on translation tasks or the skip-thought or word embedding vectors. Shi et al. (2016b); Belinkov et al. (2017) constructed regression tasks that predict labels such as voice, tense, part-of-speech tag, and morpheme from the encoder representation of the learned model in translation task.

Compared with previous work, our contributions can be summarized as follows. (1) By identifying the role of the individual units, rather than analyzing the representation as a whole, we provide more fine-grained understanding of how the representations encode informative factors in training data. (2) Rather than limiting the linguistic features within the representation to be discovered, we focus on covering concepts of fundamental building blocks of natural language (morphemes, words, and phrases) present in the training data, providing more flexible interpretation results without relying on a predefined set of concepts. (3) Our concept alignment method does not need any additional labeled data or re-training process, so it can always provide deterministic interpretation results using only the training data.

## 3 APPROACH

We focus on convolutional neural networks (CNNs), particularly their character-level variants. CNNs have shown great success on various natural language applications, including translation and sentence classification (Kalchbrenner et al., 2016; Kim et al., 2016; Zhang et al., 2015; Conneau et al., 2017). Compared to deep architectures based on fully connected layers, CNNs are natural candidates for unit-level analysis because their channel-level representations are reported to work as templates for detecting concepts (Bau et al., 2017).

Our approach for aligning natural language concepts to units is summarized as follows. We first train a CNN model for each natural language task (*e.g.* translation and classification) and retrieve training sentences that highly activate specific units. Interestingly, we discover morphemes, words and phrases that appear dominantly within these retrieved sentences, implying that those concepts have a significant impact on the activation value of the unit. Then, we find a set of concepts which attribute a lot to the unit activation by measuring activation value of each replicated candidate concept, and align them to unit.

### 3.1 TOP $K$ ACTIVATED SENTENCES PER UNIT

Once we train a CNN model for a given task, we feed again all sentences $\mathcal{S}$ in the training set to the CNN model and record their activations. Given a layer and sentence $s \in \mathcal{S}$, let $A_u^l(s)$

| Dataset | Task | Model | # of Layers | # of Units |
|---------|------|-------|-------------|------------|
| AG News | Ontology Classification | VDCNN | 4 | [64, 128, 256, 512] |
| DBpedia | Topic Classification | VDCNN | 4 | [64, 128, 256, 512] |
| Yelp Review | Polarity Classification | VDCNN | 4 | [64, 128, 256, 512] |
| WMT17' EN-DE | Translation | ByteNet | 15 | [1024] for all |
| WMT14' EN-FR | Translation | ByteNet | 15 | [1024] for all |
| WMT14' EN-CS | Translation | ByteNet | 15 | [1024] for all |
| EN-DE Europarl-v7 | Translation | ByteNet | 15 | [1024] for all |

Table 1: Datasets and model descriptions used in our analysis.

denote the activation of unit $u$ at spatial location $l$. Then, for unit $u$, we average activations over all spatial locations as $a_u(s) = \frac{1}{Z} \sum_l A_u^l(s)$, where $Z$ is a normalizer. We then retrieve top $K$ training sentences per unit with the highest mean activation $a_u$. Interestingly, some natural language patterns such as morphemes, words, phrases frequently appear in the retrieved sentences (see Figure 1), implying that those concepts might have a large attribution to the activation value of that unit.

## 3.2 CONCEPT ALIGNMENT WITH REPLICATED TEXT

We propose a simple approach for identifying the concepts as follows. For constructing candidate concepts, we parse each of top $K$ sentences with a constituency parser (Kitaev & Klein, 2018). Within the constituency-based parse tree, we define candidate concepts as all terminal and non-terminal nodes (*e.g.* from sentence *John hit the balls*, we obtain candidate concepts as {*John, hit, the, balls, the balls, hit the balls, John hit the balls*}). We also break each word into morphemes using a morphological analysis tool (Virpioja et al., 2013) and add them to candidate concepts (*e.g.* from word *balls*, we obtain morphemes {*ball, s*}). We repeat this process for every top $K$ sentence and build a set of candidate concepts for unit $u$, which is denoted as $\mathcal{C}_u = \{c_1, ..., c_N\}$, where $N$ is the number of candidate concepts of the unit.

Next, we measure how each candidate concept contributes to the unit's activation value. For normalizing the degree of an input signal to the unit activation, we create a synthetic sentence by replicating each candidate concept so that its length is identical to the average length of all training sentences (*e.g.* candidate concept *the ball* is replicated as *the ball the ball the ball...*). Replicated sentences are denoted as $\mathcal{R} = \{r_1, ..., r_N\}$, and each $r_n \in \mathcal{R}$ is forwarded to CNN, and their activation value of unit $u$ is measured as $a_u(r_n) \in \mathbb{R}$, which is averaged over $l$ entries. Finally, the degree of alignment (DoA) between a candidate concept $c_n$ and a unit $u$ is defined as follows:

$$\text{DoA}_{u,c_n} = a_u(r_n) \tag{1}$$

In short, the DoA[1] measures the extent to unit $u$'s activation is sensitive to the presence of candidate concept $c_n$. If a candidate concept $c_n$ appears in the top $K$ sentences and unit's activation value $a_u$ is responsive to $c_n$ a lot, then $\text{DoA}_{u,c_n}$ gets large, suggesting that candidate concept $c_n$ is strongly aligned to unit $u$.

Finally, for each unit $u$, we define a set of its aligned concepts $\mathcal{C}_u^* = \{c_1^*, ..., c_M^*\}$ as $M$ candidate concepts with the largest DoA values in $C_u$. Depending on how we set $M$, we can detect different numbers of concepts per unit. In this experiment, we set $M$ to 3.

## 4 EXPERIMENTS

### 4.1 THE MODEL AND THE TASK

We analyze representations learned on three classification and four translation datasets shown in Table 1. Training details for each dataset are available in Appendix B. We then focus on the representations in each encoder layer of ByteNet and convolutional layer of VDCNN, because as Mou et al. (2016) pointed out, the representation of the decoder (the output layer in the case of classification) is specialized for predicting the output of the target task rather than for learning the semantics of the input text.

---

[1] We try other metrics for DoA, but all of them induce intrinsic bias. See Appendix A for details.

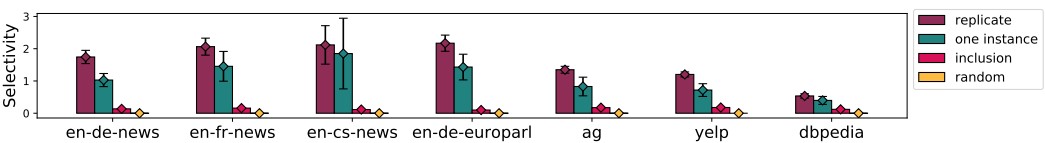

Figure 2: Mean and variance of selectivity values over all units in the learned representation for each dataset. Sentences including the concepts that our alignment method discovers always activate units significantly more than random sentences. See section 4.2 for details.

## 4.2 Evaluation of concept alignment

To quantitatively evaluate how well our approach aligns concepts, we measure how selectively each unit responds to the aligned concept. Motivated by Morcos et al. (2018), we define the **concept selectivity** of a unit $u$, to a set of concepts $\mathcal{C}_u^*$ that our alignment method detects, as follows:

$$\mathsf{Sel}_u = \frac{\mu_+ - \mu_-}{\max_{s \in \mathcal{S}} a_u(s) - \min_{s \in \mathcal{S}} a_u(s)} \tag{2}$$

where $\mathcal{S}$ denotes all sentences in training set, and $\mu_+ = \frac{1}{|\mathcal{S}_+|} \sum_{s \in \mathcal{S}_+} a_u(s)$ is the average value of unit activation when forwarding a set of sentences $\mathcal{S}_+$, which is defined as one of the following:

- *replicate*: $\mathcal{S}_+$ contains the sentences created by replicating each concept in $\mathcal{C}_u^*$. As before, the sentence length is set as the average length of all training sentences for fair comparison.

- *one instance*: $\mathcal{S}_+$ contains just one instance of each concept in $\mathcal{C}_u^*$. Thus, the input sentence length is shorter than those of others in general.

- *inclusion*: $\mathcal{S}_+$ contains the training sentences that include at least one concept in $\mathcal{C}_u^*$.

- *random*: $\mathcal{S}_+$ contains randomly sampled sentences from the training data.

In contrast, $\mu_- = \frac{1}{|\mathcal{S}_-|} \sum_{s \in \mathcal{S}_-} a_u(s)$ is the average value of unit activation when forwarding $\mathcal{S}_-$, which consists of training sentences that do *not* include any concept in $\mathcal{C}_u^*$. Intuitively, if unit $u$'s activation is highly sensitive to $\mathcal{C}_u^*$ (*i.e.* those found by our alignment method) and if it is not to other factors, then $\mathsf{Sel}_u$ gets large; otherwise, $\mathsf{Sel}_u$ is near 0.

Figure 2 shows the mean and variance of selectivity values for all units learned in each dataset for the four $\mathcal{S}_+$ categories. Consistent with our intuition, in all datasets, the mean selectivity of the *replicate* set is the highest with a significant margin, that of *one instance, inclusion* set is the runner-up, and that of the *random* set is the lowest. These results support our claims that units are selectively responsive to specific concepts and our method is successful to align such concepts to units. Moreover, the mean selectivity of the *replicate* set is higher than that of the *one instance* set, which implies that a unit's activation increases as its concepts appear more often in the input text.

## 4.3 Concept Alignment of Units

Figure 3 shows examples of the top $K$ sentences and the aligned concepts that are discovered by our method, for selected units. For each unit, we find the top $K = 10$ sentences that activate the most in several encoding layers of ByteNet and VDCNN, and select some of them (only up to five sentences are shown due to space constraints). We observe that some patterns appear frequently within the top $K$ sentences. For example, in the top $K$ sentences that activate unit 124 of 0th layer of ByteNet, the concepts of '(', ')', '-' appear in common, while the concepts of *soft, software, wi* appear frequently in the sentences for unit 19 of 1st layer of VDCNN. These results qualitatively show that individual units are selectively responsive to specific natural language concepts.

More interestingly, we discover that many units could capture specific meanings or syntactic roles beyond superficial, low-level patterns. For example, unit 690 of the 14th layer in ByteNet captures (*what, who, where*) concepts, all of which play a similar grammatical role. On the other hand, unit 224 of the 14th layer in ByteNet and unit 53 of the 0th layer in VDCNN each captures semantically similar concepts, with the ByteNet unit detecting the meaning of certainty in knowledge (*sure, know, aware*) and the VDCNN unit detecting years (*1999, 1969, 1992*). This suggests that, although we train character-level CNNs with feeding sentences as the form of discrete symbols (*i.e.* character indices), individual units could capture natural language concepts sharing a similar semantic or grammatical role. More quantitative analyses for such concepts are available in Appendix E.

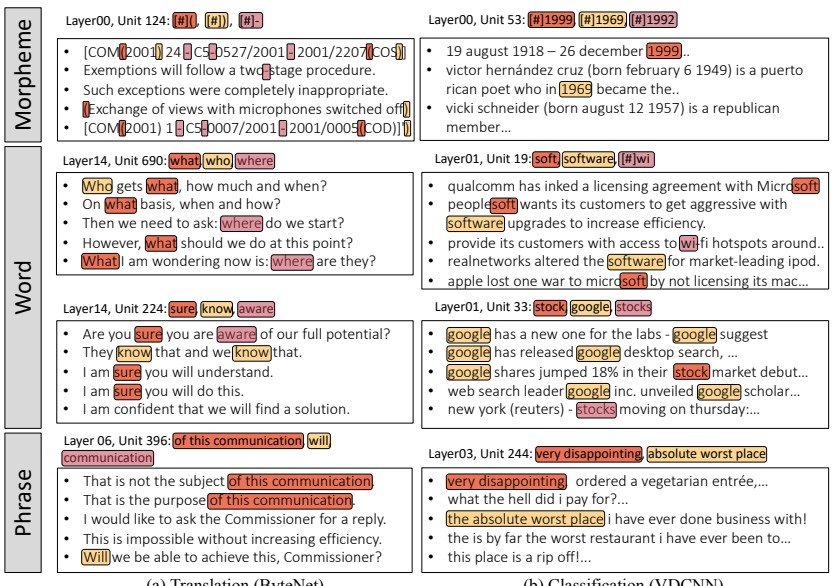

(a) Translation (ByteNet)  (b) Classification (VDCNN)

Figure 3: Examples of top activated sentences and aligned concepts to some units in several encoding layers of ByteNet and VDCNN. For each unit, concepts and their presence in top $K$ sentences are shown in the same color. [#] symbol denotes morpheme concepts. See section 4.3 for details.

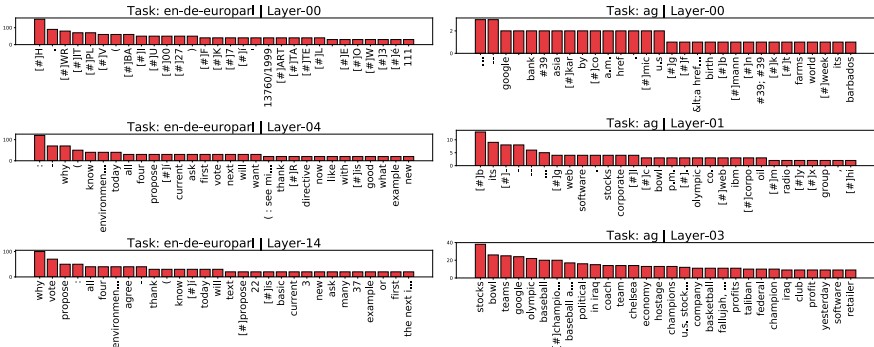

Figure 4: 30 concepts selected by the number of aligned units in three encoding layers of ByteNet learned on the Europarl translation dataset (left column) and VDCNN learned on AG-News (right column). [#] symbol denotes morpheme concepts. See section 4.4 for details.

We note that there are units that detect concepts more abstract than just morphemes, words, or phrases, and for these units, our method tends to align relevant lower-level concepts. For example, in unit 244 of the 3rd layer in VDCNN, while each aligned concept emerges only once in the top $K$ sentences, all top $K$ sentences have similar *nuances* like positive sentiments. In this case, our method does capture relevant phrase-level concepts (e.g., *very disappointing, absolute worst place*), indicating that the higher-level *nuance* (e.g., negativity) is indirectly captured.

We note that, because the number of morphemes, words, and phrases present in training corpus is usually much greater than the number of units per layer, we do not expect to always align any natural language concepts in the corpus to one of the units. Our approach thus tends to find concepts that are frequent in training data or considered as more important than others for solving the target task.

Overall, these results suggest how input sentences are represented in the hidden layers of the CNN:

- Several units in the CNN learned on NLP tasks respond selectively to specific natural language concepts, rather than getting activated in an uninterpretable way. This means that these units can serve as detectors for specific natural language concepts.

- There are units capturing syntactically or semantically related concepts, suggesting that they model the *meaning or grammatical role shared between those concepts*, as opposed to superficially modeling each natural language *symbol*.

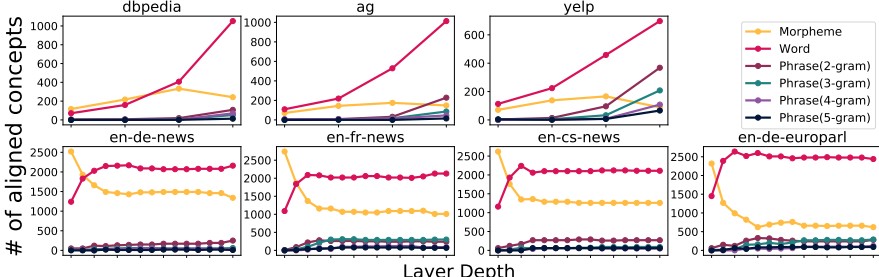

Figure 5: Aligned concepts are divided into six different levels of granularity: morphemes, words and N-gram phrases ($N = 2, 3, 4, 5$) and shown layerwise across multiple datasets and tasks. The number of units increases with layers in the classification models (*i.e.* [64, 128, 256, 512]), but in translation the number is constant (*i.e.* 1024) across all layers.

### 4.4    CONCEPT DISTRIBUTION IN LAYERS

Using the concept alignments found earlier, we can visualize how concepts are distributed across layers. Figure 4 shows the concepts of the units in the 0th, 1st, 3rd layer of VDCNN learned on AG-News dataset, and 0th, 4th, and 14th layer of the ByteNet encoder learned on English-to-German Europarl dataset with their number of aligned units. For each layer, we sort concepts in decreasing order by the number of aligned units and show 30 concepts most aligned. Recall that, since we align concepts for each unit, there are concepts aligned to multiple units simultaneously. Concept distribution for other datasets are available in Appendix G.

Overall, we find that data and task-specific concepts are likely to be aligned to many units. In AG News, since the task is to classify given sentences into following categories; *World*, *Sports*, *Business* and *Science/Tech*, concepts related to these topics commonly emerge. Similarly, we can see that units learned for Europarl dataset focus to encode some key words (*e.g. vote, propose, environment*) in the training corpus.

### 4.5    HOW DOES CONCEPT GRANULARITY EVOLVE WITH LAYER?

In computer vision tasks, visual concepts captured by units in CNN representations learned for image recognition tasks evolve with layer depths; color, texture concepts are emergent in earlier layers and more abstract concepts like parts and objects are emergent in deeper layers. To confirm that it also holds for representations learned in NLP tasks, we divide granularity of natural language concepts to the morpheme, word and $N$-gram phrase ($N = 2, 3, 4, 5$), and observe the number of units that they are aligned in different layers.

Figure 5 shows this trend, where in lower layers such as the 0th layer, fewer phrase concepts but more morphemes and words are detected. This is because we use a character-level CNN, whose receptive fields of convolution may not be large enough to detect lengthy phrases. Further, interestingly in translation cases, we observe that concepts significantly change in shallower layers (*e.g.* from the 0th to the 4th), but do not change much from middle to deeper layers (*e.g.* from the 5th to the 14th).

Thus, it remains for us to answer the following question: for the representations learned on translation datasets, *why does concept granularity not evolve much in deeper layers?* One possibility is that the capacity of the network is large enough so that the representations in the middle layers could be sufficiently informative to solve the task. To validate this hypothesis, we re-train ByteNet from scratch while varying only layer depth of the encoder and fixing other conditions. We record their BLEU scores on the validation data as shown in Figure 6. The performance of the translation model does not change much with more than six encoder layers, but it significantly drops at the models with fewer than 4 encoder layers. This trend coincides with the result from Figure 5 that the evolution of concept granularity stops around middle-to-higher layers. This shared pattern suggests that about six encoder layers are enough to encode informative factors in the given datasets to perform optimally on the translation task. In deeper models, this may suggest that the middle layer's representation may be already informative enough to encode the input text, and our result may partly coincide with that of Mou et al. (2016), which shows that representation of intermediate layers is more transferable than that of deeper layers in language tasks, unlike in computer vision where deeper layers are usually more useful and discriminative.

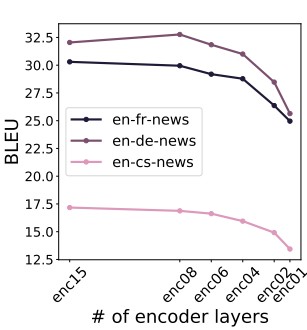
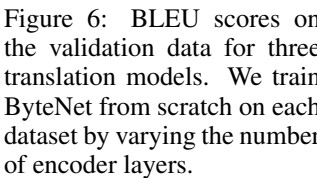

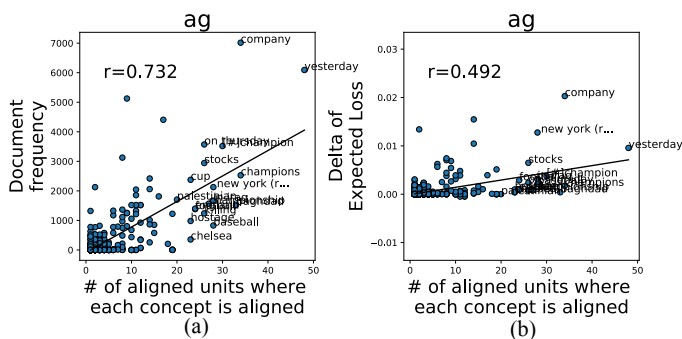

Figure 6: BLEU scores on the validation data for three translation models. We train ByteNet from scratch on each dataset by varying the number of encoder layers.

Figure 7: Correlations between the number of units per concept and (a) document frequency and (b) delta of expected loss. Pearson correlation coefficients are measured at the final layer of the AG News sentence classification task. Some concepts aligned to many units are annotated. Results on other tasks are available in Appendix F.

### 4.6 WHAT MAKES CERTAIN CONCEPTS EMERGE MORE THAN OTHERS?

We show how many units each concept is aligned per layer in Section 4.4 and Appendix G. We observe that the concepts do not appear uniformly; some concepts are aligned to many units, while others are aligned to few or even no units. Then, the following question arises: *What makes certain concepts emerge more than others?*

Two possible hypotheses may explain the emergence of dominant concepts. First, the concepts with a higher frequency in training data may be aligned to more units. Figure 7-(a) shows the correlation between the frequency of each concept in the training corpus and the number of units where each concept is aligned in the last layer of the topic classification model learned on AG News dataset.

Second, the concepts that have more influence on the objective function (expected loss) may be aligned to more units. We can measure the effect of concept $c$ on the task performance as *Delta of Expected Loss* (DEL) as follows:

$$\mathsf{DEL}(c) = \mathbb{E}_{s \in \mathcal{S}, y \in \mathcal{Y}}[\mathcal{L}(s, y)] - \mathbb{E}_{s \in \mathcal{S}, y \in \mathcal{Y}}[\mathcal{L}(\mathsf{Occ}_c(s), y)] \tag{3}$$

where $\mathcal{S}$ is a set of training sentences, and $\mathcal{Y}$ is the set of ground-truths, and $\mathcal{L}(s, y)$ is the loss function for the input sentence $s$ and label $y$. $\mathsf{Occ}_c(s)$ is an occlusion of concept $c$ in sentence $s$, where we replace concept $c$ by dummy character tokens that have no meaning. If sentence $s$ does not include concept $c$, $\mathsf{Occ}_c(s)$ equals to original sentence $s$. As a result, $\mathsf{DEL}(c)$ measures the impact of concept $c$ on the loss function, where a large positive value implies that concept $c$ has an important role for solving the target task. Figure 7-(b) shows the correlation between the DEL and the number of units per concept. The Pearson correlation coefficients for the hypothesis (a) and (b) are 0.732 / 0.492, respectively. Such high values implicate that the representations are learned for identifying the frequent concepts in the training data and important concepts for solving the target task.

## 5 CONCLUSION

We proposed a simple but highly effective concept alignment method for character-level CNNs to confirm that each unit of the hidden layers serves as detectors of natural language concepts. Using this method, we analyzed the characteristics of units with multiple datasets on classification and translation tasks. Consequently, we shed light on how deep representations capture the natural language, and how they vary with various conditions.

An interesting future direction is to extend the concept coverage from natural language to more abstract forms such as sentence structure, nuance, and tone. Another direction is to quantify the properties of individual units in other models widely used in NLP tasks. In particular, combining our definition of concepts with the attention mechanism (*e.g.* Bahdanau et al. (2015)) could be a promising direction, because it can reveal how the representations are attended by the model to capture concepts, helping us better understand the decision-making process of popular deep models.

ACKNOWLEDGMENTS

We appreciate Insu Jeon, Jaemin Cho, Sewon Min, Yunseok Jang and the anonymous reviewers for their helpful comments and discussions. This work was supported by Kakao and Kakao Brain corporations, IITP grant funded by the Korea government (MSIT) (No. 2017-0-01772) and Creative-Pioneering Researchers Program through Seoul National University. Gunhee Kim is the corresponding author.

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

# A   OTHER METRICS FOR DoA WITH BIASED ALIGNMENT RESULT

In section 3.2, we define Degree of Alignment (DoA) between concept $c_n$ and unit $u$ as activation value of unit $u$ for replication of $c_n$. We tried lots of stuff while we were working on DoA metrics, but a lot of it gives biased concept alignment result for several reasons. We here provide the things we tried and their reasons for failure.

## A.1   POINT-WISE MUTUAL INFORMATION

Point-wise Mutual Information (PMI) is a measure of association used in information theory and statistics. The PMI of a pair of samples $x$ and $y$ sampled from random variables $X$ and $Y$ quantifies the discrepancy between the probability of their coincidence as follows:

$$\text{pmi}(x, y) = \log \frac{p(x, y)}{p(x)p(y)} \tag{4}$$

We then define DoA between candidate concept $c_n$ and unit $u$ by using PMI as follow:

$$\text{DoA}_{u,c_n} = \text{pmi}(u, c_n) = \log \frac{p(u, c_n)}{p(u)p(c_n)}, \text{where} \tag{5a}$$

$$p(u) = \frac{\#[s \in \mathcal{S}, s \in \text{topK}(u)]}{\#[s \in \mathcal{S}]}, \tag{5b}$$

$$p(c_n) = \frac{\#[s \in \mathcal{S}, c_n \in s]}{\#[s \in \mathcal{S}]}, \tag{5c}$$

$$p(u, c_n) = \frac{\#[s \in \text{topK}(u), c_n \in s]}{\#[s \in \text{topK}(u)]} \tag{5d}$$

However, this metric has a bias of always preferring lengthy concepts even in earlier layers, which is not possible considering the receptive field of the convolution. Our intuition for this bias is consistent with Role & Nadif (2011), where it is a well-known problem with PMI, which is its tendency to give very high association scores to pairs involving low-frequency ones, as the denominator is small in such cases. If certain concept $c_n$ in top $K$ sentences is very lengthy, then its frequency in the corpus $p(c_n)$ would get very small, and $\text{pmi}(u, c_n)$ would be large with regardless of correlation between $u$ and $c_n$.

## A.2   CONCEPT OCCLUSION

We tested concept alignments with the following concept occlusion method. For each of the top $K$ sentences, we replace a $c_n$ by dummy character tokens which have no meaning, forward it to the model, and measure the reduction of the unit activation value. We repeat this for every candidate concept in the sentences – as a result, we can identify which candidate concept greatly reduce unit activation values. We thus define concepts aligned to each unit as the candidate concept that consistently lower the unit activation across the top $K$ sentences.

More formally, for each unit $u$, let $\mathcal{S} = \{s_1, ..., s_K\}$ be top $K$ activated sentences. Since we occlude each candidate concept in sentences, we define the set of candidate concept $\mathcal{C} = \{c_1, ..., c_N\}$, obtained from parsing each sentence in $\mathcal{S}$.

We define the **degree of alignment (DoA)** between a concept $c \in \mathcal{C}$ and a unit $u$ as:

$$\text{DoA}_{u,c_n} = \frac{1}{Z} \sum_{s \in \mathcal{S}} \mathbb{1}(c_n \in s)(a_u(s) - a_u(\text{Occ}_{c_n}(s))) \tag{6}$$

where $Z$ is a normalizing factor, and $a_u$ indicates the mean activation of unit $u$, $\text{Occ}_{c_n}(s)$ is a sentence $s$ where candidate concept $c_n$ is occluded, and $\mathbb{1}(c_n \in s)$ is an indicator of whether $c_n$ is included in the sentence $s$. In short, the DoA measures how much a candidate concept contributes to the activation of the unit's top $K$ sentences. If a candidate concept $c_n$ appears in the top $K$

sentences $\mathcal{S}$ and greatly reduces the activation of unit $u$, then $\text{DoA}_{u,c_n}$ gets large, implying that the $c_n$ is strongly aligned to unit $u$.

Unfortunately, this metric could not fairly compare the attribution of several candidate concepts. For example, consider the following two concepts $c_1 = hit$, $c_2 = hit\ the\ ball$ are included in one sentence. Occluding $c_2$ might gives relatively large decrement in unit activation value than that of $c_1$, since $c_1$ includes $c_2$. For this reason, the occlusion based metric is unnecessarily dependant of the length of concept, rather than it's attribution.

### A.3 INCLUSION SELECTIVITY

Note that *inclusion* selectivity in section 4.2 is also used as DoA. Recall that *inclusion* selectivity is calculated as equation 2. In this case, $\mu_+ = \frac{1}{|\mathcal{S}_+|} \sum_{s \in \mathcal{S}_+} a_u(s)$ is the average value of unit activation when forwarding a set of sentences $\mathcal{S}_+$, where $\mathcal{S}_+$ denotes that sentences including candidate concept $c_n$.

However, it induces a bias which is similar to section A.1. It always prefers lengthy phrases since those lengthy concepts occur few times in entire corpus. For example, assume that the activation value of unit $u$ for the sentence including specific lengthy phase is very high. If such a phrase occurs only one time over the entire corpus, $\mu_+$ is equal to the activation value of the sentence, which is relatively very high than $\mu_+$ for other candidate concepts. This error could be alleviated on a very large corpus where every candidate concept occurs enough in the corpus so that estimation of $\mu_+$ get relatively accurate, which is practically not possible.

### A.4 COMPUTING DOA VALUES WITHOUT REPLICATION

In Section 3.2, we replicate each candidate concept into the input sentence for computing DoA in Eq.(1). Since each unit works as a concept detector whose activation value increases with the length of the input sentence (Section 4.2), it is essential to normalize the length of input for fair comparison of DoA values between the concepts that have different lengths one another. Without the length-normalization (*i.e.* each input sentence consists of just one instance of the candidate concept), the DoA metric has a bias to prefer lengthy concepts (*e.g.* phrases) because they typically have more signals that affect the unit activation than short candidate concepts (*e.g.* single words).

## B TRAINING DETAILS

In this work, we trained a ByteNet for the translation tasks and a VDCNN for the classification tasks, both to analyze properties of representations for language. Training details are as follows.

### B.1 BYTENET

We trained a ByteNet on the translation tasks, in particular on the WMT'17 English-to-German Europarl dataset, the English-to-German news dataset, WMT'16 English-to-French, English-to-Czech news dataset. We used the same model architecture and hyperparameters for both datasets. We set the batch size to 8 and the learning rate to 0.001. The parameters were optimized with Adam (Kingma & Ba, 2015) for 5 epochs, and early stopping was actively used for finding parameters that generalize well. Our code is based on a TensorFlow (Abadi et al., 2015) implementation of ByteNet found in `https://github.com/paarthneekhara/byteNet-tensorflow`.

### B.2 VERY DEEP CNN (VDCNN)

We trained a VDCNN for classification tasks, in particular on the AG News dataset, the binarized version of the Yelp Reviews dataset, and DBpedia ontology dataset. For each task, we used 1 temporal convolutional layer, 4 convolutional blocks with each convolutional layer having a filter width of 3. In our experiments, we analyze representations of each convolutional block layer. The number of units in each layer representation is 64, 128, 256, 512 respectively. We set the batch size to 64 and the learning rate to 0.01. The parameters are optimized using SGD optimizer for 50 epochs, and early stopping is actively used. For each of the AG News, Yelp Reviews and DBpedia datasets, a VD-

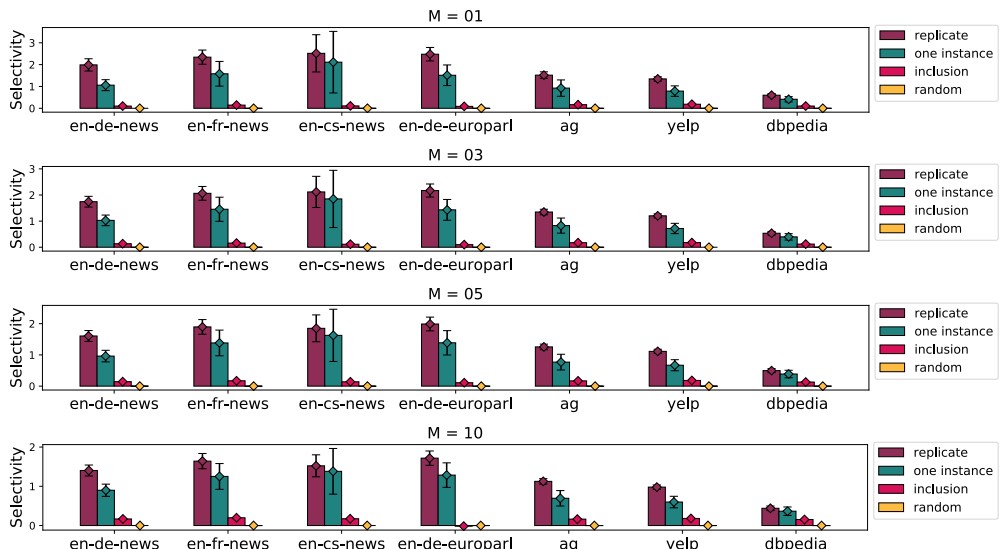

Figure 8: Mean and variance of selectivity values for different $M = [1, 3, 5, 10]$, where $M$ is the number of selected concepts per unit. In all settings, the selectivity of *replicate* ones is the highest, that of *one instance* ones is runner-up, and that of *random* is the lowest near 0.

CNN was learned with the same structure and hyperparameters. Our code is based on a TensorFlow implementation of VDCNN found in `https://github.com/zonetrooper32/VDCNN`.

## C  VARIANTS OF ALIGNMENT WITH DIFFERENT M VALUES

In Section 3.2, we set $M = 3$. Although $M$ is used as a threshold to set how many concepts per unit are considered, different $M$ values have little influence on quantitative results such as selectivity in Section 4.2. Figure 8 shows the mean and variance of selectivity values with different $M = [1, 3, 5, 10]$, where there is little variants in the overall trend; the sensitivity of the *replicate* set is the highest, and that of *one instance* is runner-up, and that of *random* is the lowest.

## D  NON-INTERPRETABLE UNITS

Whereas some units are sensitive to specific natural language concepts as shown in Section 4.3, other unites are not sensitive to any concepts at all. We call such units as *non-interpretable units*, which deserve to be explored. We first define the unit interpretability for unit $u$ as follows:

$$\text{Interpretability}(u) = \begin{cases} 1 & \text{if } \max_{s \in \mathcal{S}}\{a_u(s)\} < \max_{i=1,\dots,N}\{r_i\} \\ 0 & \text{otherwise} \end{cases} \tag{7}$$

where $\mathcal{S}$ is the set of training sentences, $a_u(s)$ is the activation value of unit $u$, and $r_i$ is the activation value of the sentence that is made up of replicating concept $c_i$. We define unit $u$ as interpretable when its $\text{Interpretability}(u)$ equals to 1, and otherwise as non-interpretable. The intuition is that if a replicated sentence that is composed of only one concept has a less activation value than the top-activated sentences, the unit is not sensitive to the concept compared to a sequence of different words.

Figure 9 shows the ratio of the interpretable units in each layer on several datasets. We observe that more than 90% of units are interpretable across all layers and all datasets.

Figure 10 illustrates some examples of non-interpretable units with their top five activated sentences and their concepts. Unlike Figure 3, the aligned concepts do not appear frequently over top-activated sentences. This result is obvious given that the concepts have little influence on unit activation. There are several reasons why non-interpretable units appear. One possibility is that several units

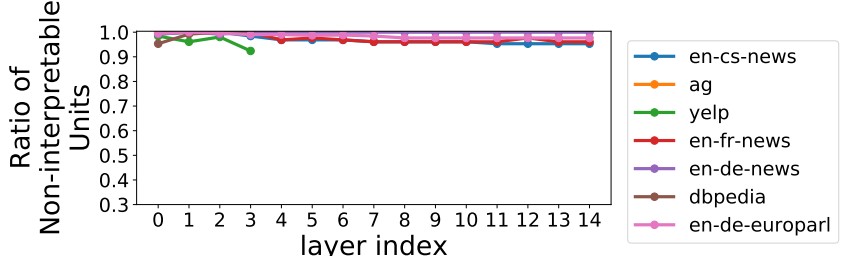

Figure 9: Ratio of interpretable units in layer-wise. Across all datasets, there are more than 90% of units are interpretable. See Section D for more details.

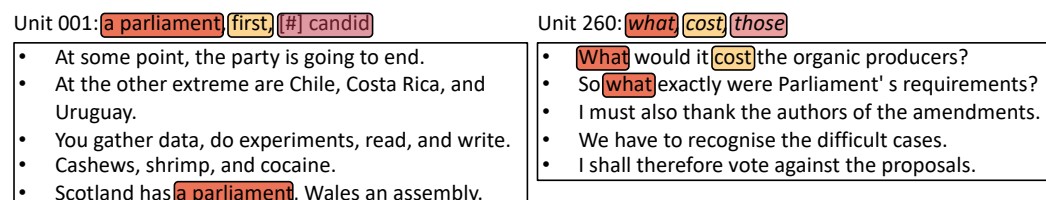

Figure 10: Examples of non-interpretable units, their concepts and top 5 activated sentences. Units are from representations learned on english-to-french translation dataset. See section D for details.

align concepts that are out of natural language form. For example, in unit 001 in the left of Figure 10, we discover that sentence structure involves many *commas* in top activated sentences. Since we limit the candidate concepts to only the form of morpheme, word and phrase, such punctuation concepts are hard to be detected. Another possibility is that some units may be so-called *dead units* that are not sensitive to any concept at all. For example, unit 260 in the right of Figure 10 has no pattern that appears consistently in top activated sentences.

## E    CONCEPT CLUSTERS

We introduce some units whose concepts have the shared meaning in Section 4.3. We here refer *concept cluster* to the concepts that are aligned to the same unit and have similar semantics or grammatical roles. We analyze how clusters are formed in the units and how they vary with the target task and layer depth.

### E.1    CONCEPT CLUSTERS BY TARGET TASKS

Figure 11 illustrates some concept clusters of units in the final layer learned on each task. Top and left dendrograms of each figure show hierarchical clustering results of 30 concepts aligned with the largest number of units. We use clustering algorithm of Müllner (2011); we define the distance between two concepts as the Euclidean distance of their vector space embedding. We use fastText (Bojanowski et al., 2017) pretrained on Wikipedia dataset to project each concept into the vector space. Since fastText is a character-level $N$-gram based word embedding, we can universally obtain the embedding for morphemes as well as words or phrases. For phrase embedding, we split it to words, project each of them and average their embeddings. The distance between two clusters is defined as the distance between their centroids.

Each central heat map represents the number of times each concept pair is aligned to the same unit. Since the concepts in the x, y-axes are ordered by the clustering result, if the diagonal blocks (concept clusters) emerge more strongly, the concepts in the same unit are more likely to have the similar meanings.

In Figure 11, the units learned in the classification tasks tend to have stronger concept clusters than those learned in the translation tasks. Particularly, the concept clusters are highly evident in the units learned in DBpedia and AG News dataset. Our intuition is that units might have more benefits to

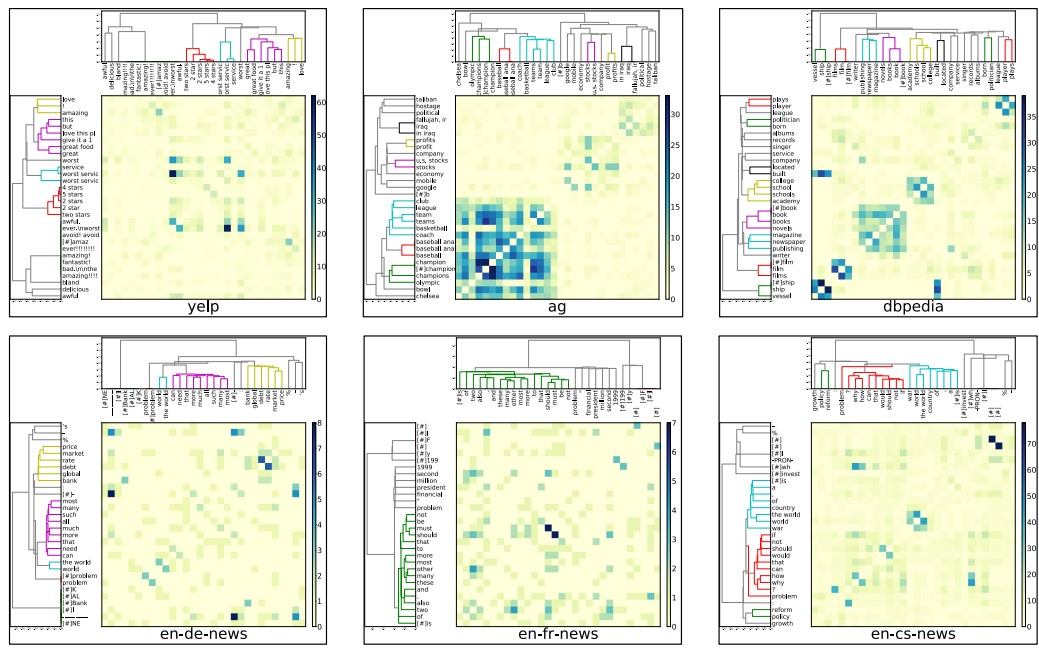

Figure 11: Concept clusters of the last layer representations learned in each task. The more distinct the diagonal blocks are, the stronger the tendency that concepts aligned to the same unit share similar meanings or semantics. See Appendix E for details.

solve the task by clustering similar concepts in the classification than the translation. That is, in the classification, input sentences that have the similar concepts tend to belong to the same class label, while in the translation, different concepts should be translated to different words or phrases even if they have similar meanings in general.

### E.2 CONCEPT CLUSTERS BY LAYERS

We analyze how concept clusters change by layer in each task. We compute the averaged pairwise distance between the concepts in each layer. We project each concept to the vector space using the three pretrained embeddings: (1) Glove (Pennington et al., 2014), (2) ConceptNet (Speer et al., 2017), (3) fastText. Glove and fastText embeddings are pretrained on Wikipedia dataset, and ConcpetNet is pretrained based on the ConceptNet graph structure.

Figure 12 shows the averaged pairwise distances in each layer. In all tasks, there is a tendency that the concepts in the same unit become closer in the vector space as the layer goes deeper. It indicates that individual units in earlier layers tend to capture more basic text patterns or symbols, while units in deeper layers capture more abstract semantics.

## F  WHAT MAKES CERTAIN CONCEPTS EMERGE MORE THAN OTHERS?: OTHER DATASETS

We investigate why certain concepts emerge more than others at Section 4.6 when the ByteNet is trained on English-to-French news dataset. Here, Figure 14 shows more results in other datasets. Consistent with our intuition, in all datasets, both the document frequency and the delta of expected loss are closely related to the number of units per concept. It concludes that the representations are learned for identifying not only the frequent concepts in the training set and but also the important concepts for solving the target task.

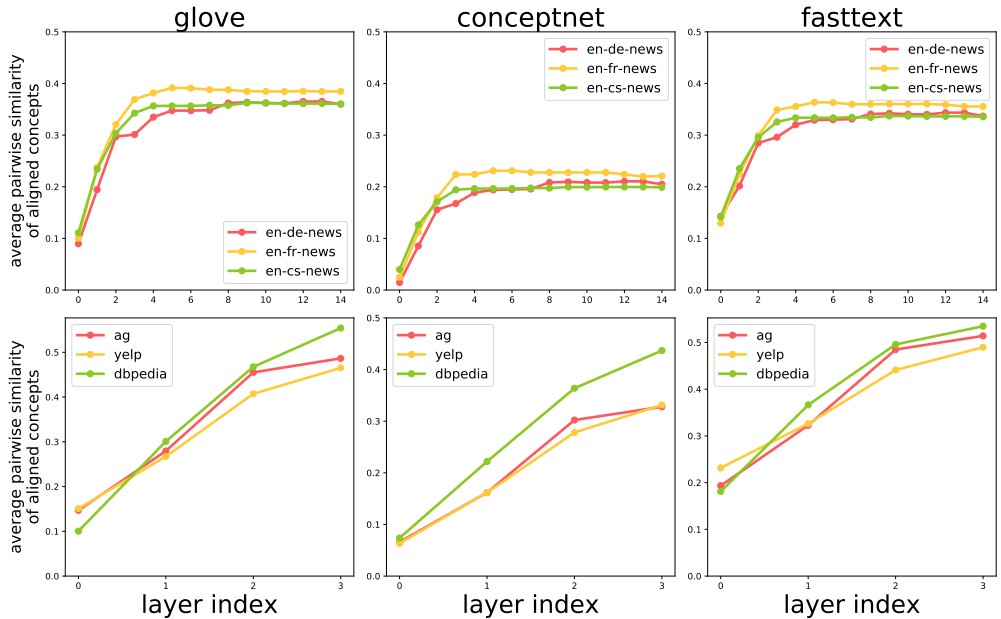

Figure 12: Averaged pairwise distances of concepts in each layer per task. Top and bottom row show the concept distances of translation models and classification models, respectively. For projecting concepts into the embedding space, we use three pretrained embedding: Glove, ConceptNet, FastText. See Appendix E.2 for more details.

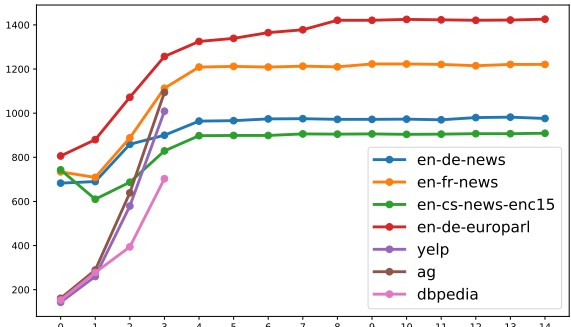

Figure 13: The number of unique concepts in each layer. It increases with the layer depth, which implies that the units in a deeper layer represent more diverse concepts.

## G   CONCEPT DISTRIBUTION IN LAYERS FOR OTHER DATASETS

In section 4.4, we visualized how concepts are distributed across layers, where the model is trained on AG News dataset and English-to-German Europarl dataset. Here, Figure 15 shows concept distribution in other datasets noted in Table 1.

In the classification tasks, we expect to find more concepts that are directly related to predicting the output label, as opposed to the translation tasks where the representations may have to include information on most of the words for an accurate translation. While our goal is not to relate each concept to one of the labels, we find several concepts that are more predictive to a particular label than others.

Consistent with section 4.4, there are data-specific and task-specific concepts aligned in each layer; *i.e.* {*worst, 2 stars, awful*} at Yelp Review, {*film, ship, school*} at DBpedia, and some key words at translation datasets. Note that Yelp Review and DBpedia is a classification dataset, where the model is required to predict the polarity (*i.e.* +1 or -1) or ontology (*i.e.* Company, Educational

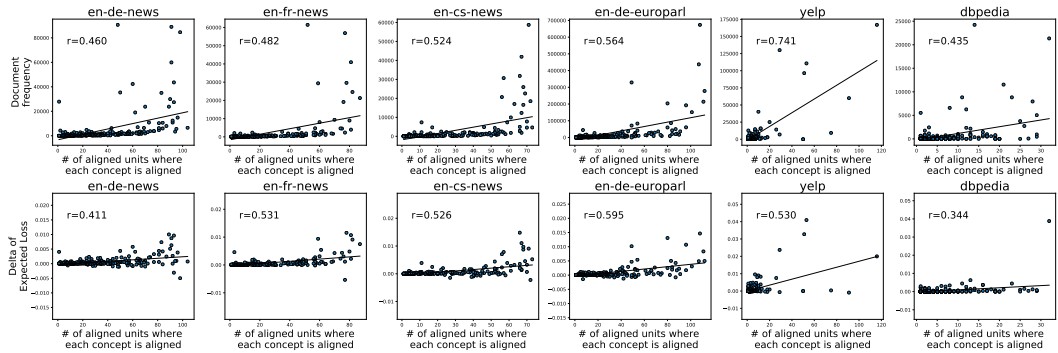

Figure 14: The Pearson correlations between the number of units per concept and (i) document frequency (top row), (ii) delta of expected loss (bottom row). They are measured at the final layer representation.

Institution, Artist, Athlete, Officeholder, Mean of Transportation, Building, Natural Place, Village, Animal, Plant, Album, Film, Written Work) for given sentence in supervised setting.

## H   MULTIPLE OCCURRENCES OF EACH CONCEPT AT DIFFERENT LAYERS

Figure 16 shows the number of occurrences of each concept at different layers. We count how many times each concept appears across all layers and sort them in decreasing order. We select two concepts in the translation model and seven concepts in the classification model, as to their number of occurrences. For example, since there are 15 encoder layers in the ByteNet translation model, we select 30 concepts in total. Although task and data specific concepts emerge at different layers, there is no strong pattern between the concepts and their occurrences at multiple layers.

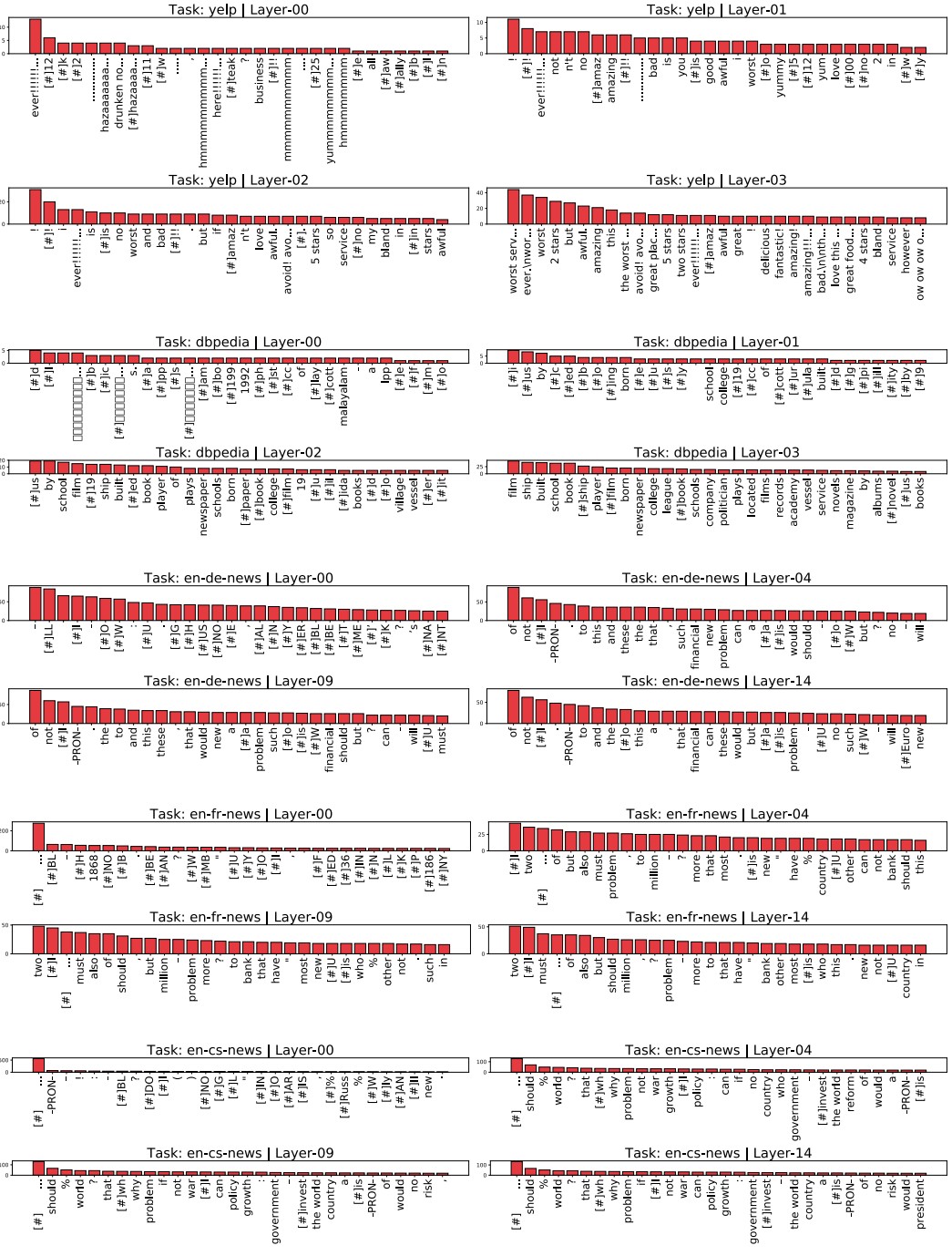

Figure 15: 30 concepts selected by the number of aligned units in four encoding layers in VD-CNN learned on Yelp Review dataset and DBpedia ontology dataset, and ByteNet learned on the English-to-German, English-to-French, and English-to-Czech parallel corpus. [#] symbol denotes morpheme concept.

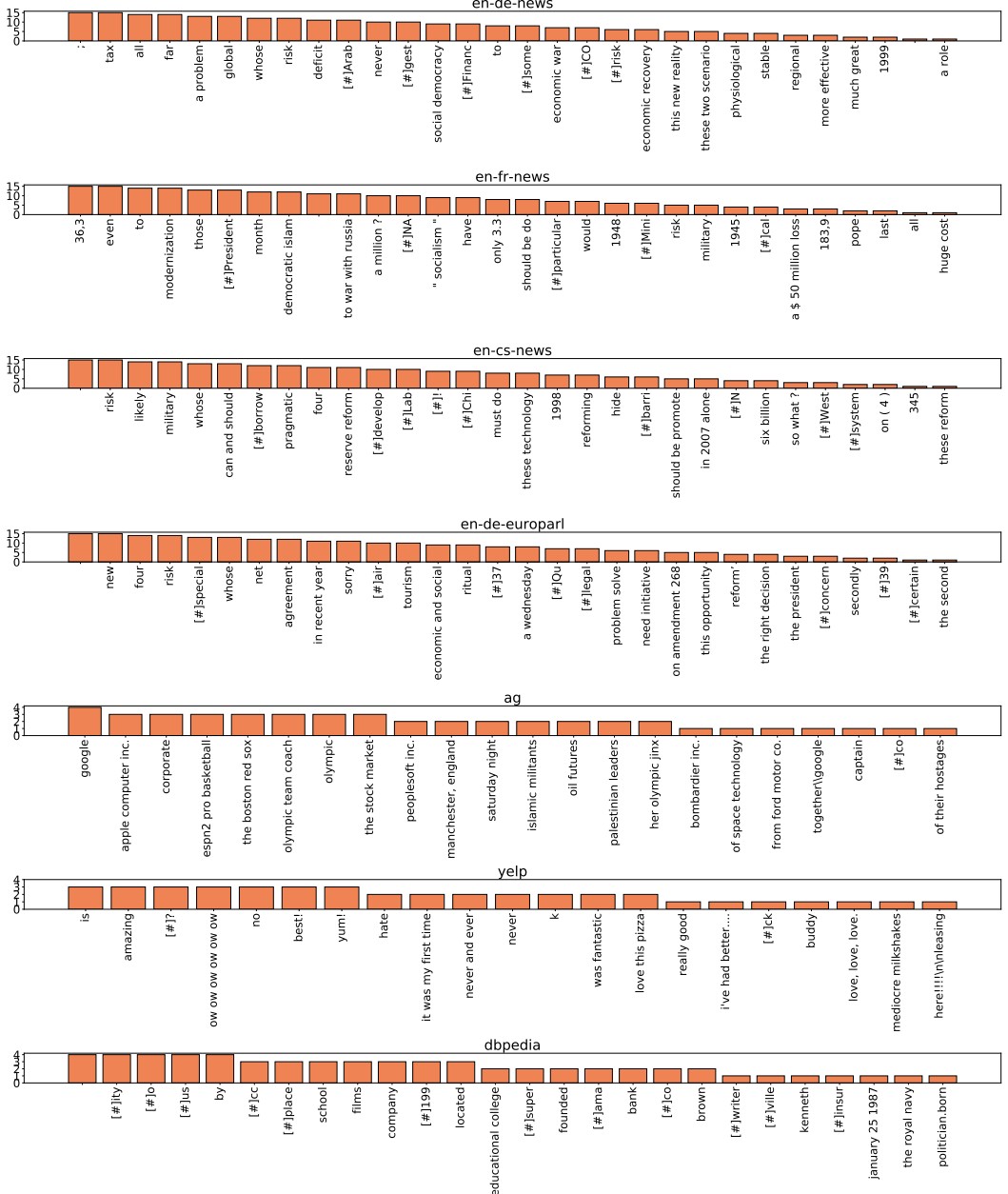

Figure 16: Aligned concepts per each task and their number of occurrences over multiple layers. See Appendix H for more details.

