# OpenReview forum: "Discovery of Natural Language Concepts in Individual Units of CNNs"
_ICLR.cc/2019/Conference_

### Official Review · AnonReviewer1 · 2018-11-02
**This paper proposes an interpretation to the activation values of hidden layer units of convolutional neural networks trained on language tasks, aligning those units with natural language concepts. The work is novel and interesting to the NLP community.**

**Rating:** 6
**Confidence:** 3

**Review:**

The paper is well written and structured, presenting the problem clearly and accurately. It contains considerable relevant references and enough background knowledge. It nicely motivates the proposed approach, locates the contributions in the state-of-the-art and reviews related work. It is also very honest in terms of how it differs on the technical level from existing approaches.
The paper presents interesting and novel findings to further state-of-the-art’s understanding on how language concepts are represented in the intermediate layers of deep convolutional neural networks, showing that channels in convolutional representations are selectively sensitive to specific natural language concepts. It also nicely discusses how concepts granularity evolves with layers’ deepness in the case of natural language tasks.
What I am missing, however, is an empirical study of concepts coverage over multiple layers, studying the multiple occurrences of single concepts at different layers, and a deeper dive on the rather noisy elements of natural language and the layers’ activation dynamics towards such elements.
Overall, however, the ideas presented in the paper are interesting and original, and the experimental section is convincing. My recommendation is to accept this submission.

---

> ### Author Response · Authors · 2018-11-27
> **Response to Reviewer 1**
>
>
> We thank Reviewer 1 for positive and constructive review. Please see our revisions in blue font to check how our paper is updated.
>
> 1. Concepts coverage over multiple layers
> ===================================
> We plot the number of unique concepts per layer in Figure 13. In all datasets, the number of unique concepts increases with the layer depth, which implies that the units in a deeper layer represent more diverse concepts.
>
>
> 2. Multiple occurrences of each concept at different layers
> ===================================
> We add Figure 16 to Appendix H to show how many layers each concept appears. Although task and data specific concepts emerge at different layers, there is no strong pattern between the concepts and their occurrences at multiple layers.
>
>
> 3. The layers’ activation dynamics towards noisy elements
> ===================================
> It is an interesting suggestion to investigate how unit activations vary with noisy elements of natural language such as synthetic adversarial examples or natural noise (Belinkov et al.[1]) that could attack the model. Since we discover some units that capture the abstract semantics rather than low-level text patterns in Section 4.2, we expect that those units will be not sensitive to such noisy transformation of the concepts. More thorough analysis for this topic will be one of our emergent future works.
>
> References
> ===================================
> [1] Yonatan Belinkov et al., Synthetic and Natural Noise Both Break Neural Machine Translation (ICLR 2018)

---

### Official Review · AnonReviewer3 · 2018-11-02
**Interesting results on an important problem, but insufficient analysis and evaluation**

**Rating:** 6
**Confidence:** 4

**Review:**

========== Edit following authors' response  ==========

Thank you for your detailed response and updated version. I think the new revision is significantly improved, mainly in more quantitative analyses and details in several places. I have updated my evaluation accordingly.

See a few more points below.

1. Thank you for clarifying your definition of concepts. I still think that the word "concept" has a strong semantic connotation, while the linguistic elements your analyses capture may do other things. The results in appendix E do show that some semantic clusters arise. It's especially interesting to see the blocks in some of the heat maps, where similar "concepts" are clustered together (like the sports terms in AG); consider commenting on this.

2. The new quantitative analyses are helpful. One other suggestion that I mentioned before is to connect detected concepts to external resources like WordNet or ConceptNet. That would help show that "concepts" are indeed semantic objects.

3. The motivation for replicating as normalizing for length does make sense, although the input would still be unnatural. The comparison to "one instance" is helpful, but it's interesting that the differences between it and replication in figure 2 are not large. It would be good to show results that substantiate your assumption that without replication there will be a bias towards lengthy concepts. Does "one instance" detect more lengthy concepts than replication?

4. The results on frequency and loss difference in 4.5 are very interesting. There is another angle to consider frequency: words that appear frequently often carry less semantic content (e.g. function words), so one might conjecture that they would require less units. It may be interesting to look at which concepts are detected at each frequency bin.

5. Minor points: section 2.2 still mentions "regression" where it should be "classification".

6. A few remaining grammar issues:
- "one concept has a less activation value.." - rephrase
- end of section 3.3: "this experiments" -> "these experiments"


========== Original review follows ==========

Summary:
=======
This paper analyzes individual units in CNN models for text classification and translation tasks. It defines a measure of sensitivity for a unit and evaluates how sensitive each unit is to "concepts" in the input text, where concepts are morphemes, words, and phrases. The analysis shows that some units seem to learn semantic concepts, while others capture linguistic elements that are frequent or relevant for the end task. Layer-wise results show some correspondence between layer depth and linguistic element size.

The paper studies an important question that is relatively under-studies in NLP compared to the computer vision community. The motivation for the work is quite convincing.
I found some of the results and analysis interesting, but overall felt that the work can be made much stronger by more quantitative evaluations. I am also worried that the notion of "concept" is misleading here. See below for this and other comments. I am willing to reconsider my evaluation pending response to the below issues.

Main comments:
=============
1. Concepts:
- morphemes, words, and phrases - are these "concepts"? They are indeed "fundamental building blocks of natural language" (2.2), but "concepts" has a more semantic connotation that I'm not sure these units target at.
- Some of the results do suggest that units learn concepts, as the analysis in 4.2 shows a "unit detecting the meaning of certainty in knowledge" and later units that have similar sentiments. It would be informative to quantify this in some way, for example by matching detected concepts to WordNet synsets, sentiment lexicons, etc., or else tagging and classifying them with various NLP tools. This could also reveal if units learn more syntactic or semantic concepts, and so on.
2. Generally, many of the analyses in the paper are qualitative and on a small scale. The results will be more convincing with more automatic aggregate measures.
3. The structure of the paper is confusing. Section3 starts with the approach but then mentions datasets and tasks (3.1). Section 4 is titled experiments, but section 4.1 starts with defining the concept selectivity. I would suggest reorganizing sections 3 and 4, such that section 3 describes all the methods and metrics, while dataset-specific parts are moved to section 4.
4. section 3.2 should provide more details on the sentence representation and how its obtained in the CNN models. A mathematical derivation and/or figure could be helpful. It is also not clear to me what's the motivation for mean-pooling over the l entries of the vector.
5. section 3.3: the use of replicated text for "concept alignment" is puzzling. This is not a natural input to the model, and I think more justification and motivation åre needed for this issue, as well as perhaps comparison with other approaches.
6. I found section 4.4 very interesting. It shows some intuitive results of larger linguistic elements learned at higher layers, but then some results that do not show such a trend. Then, hypothesizing that the middle layers are sufficient AND validating the hypothesis by retraining the model is excellent. It's a very nice demonstration that the analysis can lead to model improvements.
7. Figure 2 seems to be almost caused by construction of the different options for S_+. Is it surprising that the replicate set has the highest sensitivity? Is there a better control setup than comparing with a random set?
8. One concern that I have is the effect of confounding factors like frequency on the results. The papers occasionally attributes importance to concepts (e.g. in 4.2), but I wonder if instead we may be seeing more frequent words. Controlling for the effect of frequency would be useful.


Minor comments:
==============
- Section 2.2, first paragraph: regression should be changed to classification
- The related work is generally relevant, although one could mention a few other papers that analyzed individual neurons in NLP tasks [1, 2]
- section 4.1: the random set may perhaps be denoted by something more neutral, not S_+ as the replicate and inclusion sets.
- section 4.3, last paragraph: listing examples showing that units in Europarl focus on key words would be good.
- Figure 5, y axis label: should this be number of units instead of concepts?
- Appendix A has several interesting points but there is no reference to them from the main paper.


Writing, grammar, etc.:
======================
- Introduction: among them - who is them?
- 2.1: motivated from -> motivated by; In computer vision community -> In the computer vision community
- 2.1: quantifying characteristics of representations in layer-wise -> rephrase
- 3.2: dimension of sentence -> dimension of the/a sentence
- 4.1: to which -> remove "which"
- 4.2: in the several encoding layer -> in several encoding layers
- 4.3: aliged -> aligned
- Capitalize titles in references
- A.2: with following -> with the following; how much candidate -> how much a candidate; consider following -> consider the following
- A.3: induces similar bias -> induces a bias; such phrase -> such a phrase; on very -> on a very
- C: where model -> where the model; In consistent -> Consistent; where model -> where the model


References
==========
[1] Qian et al., Analyzing linguistic knowledge in sequential model of sentence
[2] Shi et al., Why Neural Translations are the Right Length

---

> ### Author Response · Authors · 2018-11-27
> **Response to Reviewer 3 (part 1)**
>
>
> We thank Reviewer 3 for positive and constructive review. Please see blue fonts in the newly uploaded draft to check how our paper is updated.
>
> 1. Concepts
> ===================================
> (1) We agree that the term ‘concept’ could be ambiguous. Nonetheless, we use the term ‘concept’, following the related work for interpretability [1-4], where the ‘units’ and ‘concepts’ are typically used to refer to the channels of hidden layers and the detected semantic parts of the input information (eg, wheels, cars, legs as visual concepts), respectively. In our work on natural language, the ‘concepts’ in previous work should correspond to morphemes, words, and phrases, which form the fundamental building blocks of natural language. Please also note that we define ‘natural language concept’ in Section 1 instead of ‘concept’ alone for less confusion.
>
>
> (2) We define a “concept cluster” as a set of concepts that are aligned to the same unit and have similar semantics or grammatical roles. We add what concept clusters emerge per task to Appendix E.1. We observe that such concept clusters appear more strongly in classification tasks rather than translation tasks. Also, we investigate how concept clusters vary with layer depth and discuss the detailed results in Appendix E.2, where we discover that units in deeper layers tend to form clusters more strongly than units in earlier layers. Please refer to Appendix E for more results.
>
>
> 2. Analyses are qualitative and in a small scale
>  ===================================
> Given that we use two state-of-the-art models on seven benchmark datasets, our experiments are large-scale, although some analyses are done qualitatively in small-scale as Reviewer pointed out.
> Therefore, we add more quantitative and thorough analyses as follows.
> (1) Ratios of interpretable/non-interpretable units across layers for multiple tasks and datasets (Appendix D).
> (2) Quantitative measures of concept clusters across layers for multiple tasks and datasets (Appendix E).
> (3) Correlation coefficients of possible hypotheses on why certain units emerge (i.e. document frequency and delta of expected loss) for multiple tasks and datasets (Section 4.5 and Appendix F).
> (4) Selectivity variation for different M values = [1,3,5,10] (Appendix C).
> (5) The number of unique concepts aligned to each layer for multiple tasks and datasets. (Figure 13)
>
> 3. Paper structure
>  ===================================
> Per Reviewer 3’s suggestion, we will move [The Model and the Task] Section to 4.1 in the camera-ready version.
>
>
> 4. Sentence representation
>  ===================================
> (1) We clarify Section 3.2 as Reviewer 3 suggested. Please refer to blue fonts in Section 3.2
> (2) The idea of mean-pooling over all spatial locations is motivated by Zhou et al. [4]. The only difference is that [4] uses the addition pooling because the input set is fixed-length images, whereas we use the mean pooling because the input is variable-length sentences.

---

> > ### Author Response · Authors · 2018-11-27
> > **Response to Reviewer 3 (part 2)**
> >
> >
> > 5. Concept replication
> >  ===================================
> > The main reason that we replicate each concept into a fixed-length sentence is to normalize the degree of the input signal to the unit activation. We clarify this point in Section 3.3. Without such normalization (e.g. a single instance of a candidate concept as input, as Reviewer 2 suggested), the DoA metric has a bias to prefer a lengthy concept. Please refer to Appendix A.4 for comparison with 'one instance' method.
> >
> >
> > 6. Section 4.4
> >  ===================================
> > We thank Reviewer 3 for acknowledging the significance of results in section 4.4.
> >
> >
> > 7. Sensitivity of replicate setting
> >  ===================================
> > We add a ‘one instance’ option to the comparison of selectivity (Fig. 2). The results show that the mean selectivity of the ‘replicate’ set is higher than that of the ‘one instance’ set, which implies that a unit's activation increases as its concepts appear more often in the input text. One of our main contributions is the discovery of the units that are selectively responsive to specific natural language concepts and “it is quantitatively verified” in Fig. 2.
> >
> >
> > 8. Factors that affect concept alignment
> >  ===================================
> > It is an interesting question why certain concepts emerge more than others. We experiment some factors that may affect concept alignment, and add results to Section 4.5 and Appendix F. We investigate the following two hypotheses: (i) The concepts with higher frequency in training data are aligned to more units (as Reviewer 3 suggested). (ii) Concepts that have more influence on the objective function (expected loss) are aligned to more units. For the concepts in the final layer of translation model, we measure the Pearson correlation coefficient between [# of aligned units per concept] and the factor (i) and (ii), and obtain 0.482 / 0.531, respectively. These results make a lot of sense in that the learned representation focuses more on identifying both frequent concepts and important concepts for solving the target task. Yet, we are not sure that we should directly “control” the effect of frequency, because it is quite unnatural and non-trivial to manipulate the training data to change the frequency of a specific concept.
> >
> >
> > 9. Minor comments from Reviewer 3
> > ===================================
> > (1) We update Section 2.2, related work, Section 4.1 and Section 4.3 as Reviewer 3 suggested. Please see the blue fonts.
> > (2) Fig. 5: We thank Reviewer 3 for correcting the typo. The y-axis of Fig. 5 is “the number of aligned concepts” in each layer. For example, the plot on the top left dbpedia shows that more than 100 morpheme concepts are aligned across all units of the 0-th layer. We also update the caption of Fig. 5 for clarification.
> > (3) Appendix A: We add reference to Appendix A in footnote of Section 3.3 of the revised paper.
> > (4) Notation of set of ‘random’ sentences: we will modify notation of random set for less confusing in the camera-ready version.
> >
> > 10. Writing and grammar
> > ===================================
> > We sincerely thank Reviewer 3 for thorough proofreading. We correct all the typos.
> >
> > Reference
> > ===================================
> > [1] Bolei Zhou et al., Revisiting the Importance of Individual Units in CNNs via Ablation (arXiv:1806.02891, 2018)
> > [2] David Bau et al., Network Dissection: Quantifying Interpretability of Deep Visual Representations (CVPR 2017)
> > [3] Ruth Fong et al., Net2Vec: Quantifying and Explaining how Concepts are Encoded by Filters in Deep Neural Networks (CVPR 2018)
> > [4] Bolei Zhou et al., Object Detectors Emerge In Deep Scene CNNs (ICLR 2015)

---

> ### Author Response · Authors · 2018-12-14
> **Response to Reviewer 3 (for post-rebuttal comments)**
>
>
> We are deeply grateful to reviewer3 for thoughtful post-rebuttal suggestions. We will clarify terminology, add more analyses and modify the figures accordingly. For example, we will match the detected concepts with those in WordNet (ConceptNet) tree and update Fig 7 and Fig 14 to show which concepts are detected at each bin.

---

### Official Review · AnonReviewer2 · 2018-11-03
**Solid paper with interesting insights - left with some questions**

**Rating:** 6
**Confidence:** 4

**Review:**

This paper describes a method for identifying linguistic components ("concepts") to which individual units of convolutional networks are sensitive, by selecting the sentences that most activate the given unit and then quantifying the activation of those units in response to subparts of those sentences that have been isolated and repeated. The paper reports analyses of the sensitivities of different units as well as the evolution of sensitivity across network layers, finding interesting patterns of sensitivity to specific words as well as higher-level categories.

I think this paper provides some useful insights into the specialization of hidden layer units in these networks.  There are some places where I think the analysis could go deeper / some questions that I'm left with (see comments below), but on the whole I think that the paper sheds useful light on the finer-grained picture of what these models learn internally. I like the fact that the analysis is able to identify a lack of substantial change between middle and deeper layers of the translation model, which inspires a prediction - subsequently borne out - that decreasing the number of layers will not substantially reduce task performance.

The paper is overall written pretty clearly (though some of the questions below could likely be attributed to sub-optimal clarity), and to my knowledge the analyses and insights that it contributes are original. Overall, I think this is a solid paper with some interesting contributions to neural network interpretability.

Comments/questions:

-I'm wondering about the importance of repeating the “concepts” to reach the average sentence length. Do the units not respond adequately with just one instance of the concept (eg "the ball" rather than "the ball the ball the ball")? What is the contribution of repetition alone?

-Did you experiment with any other values for M (number of aligned candidate concepts per unit)? It seems that this is a non-trivial modeling decision, as it has bearing on the interesting question of how broadly selective a unit is.

-You give examples of units that have interpretable sensitivity patterns - can you give a sense of what proportion of units do *not* respond in an interpretable way, based on your analysis?

-What exactly is plotted on the y-axis of Figure 5? Is it number of units, or number of concepts? How does it pool over different instances of a category (different morphemes, different words, etc)? What is the relationship between that measure and the number of distinct words/morphemes etc that produce sensitivity?

-I'm interested in the units that cluster members of certain syntactic and semantic categories, and it would be nice to be able to get a broader sense of the scope of these sensitivities. What examples of these categories are captured? Is it clear why certain categories are selected over others? Are they obviously the most optimal categories for task performance?

-p7 typo: "morhpeme"

---

> ### Author Response · Authors · 2018-11-27
> **Response to Reviewer 2**
>
>
> We thank Reviewer 2 for positive and constructive review. Please see blue fonts in the newly uploaded draft to check how our paper is updated.
>
> 1.  Replicating concepts
> ===================================
> The main reason that we replicate each concept into a fixed-length sentence is to normalize the degree of the input signal to the unit activation. Without such normalization (e.g. a single instance of a candidate concept as input, as Reviewer 2 suggested), the DoA metric has a bias to prefer a lengthy concept. We clarify this point at Section 3.3, and present detailed discussion in Appendix A.4.
>
>
> 2. M values
> ===================================
> The M value is used as a threshold to set how many concepts per unit are considered for later analyses. We observe that the overall trend in our quantitative results does not change much with M. As an example, we add Fig.8 to Appendix C, which shows the trend of selectivity values is stable across different M= [1,3,5,10].
>
>
> 3. Non-interpretable units
> ===================================
> It is a highly interesting suggestion to investigate non-interpretable units as well as interpretable ones. We add one approximate method to quantify the non-interpretability of unit to Appendix D in the revised paper.
> We define a unit as non-interpretable, if the activation value of its top-activated sentence is higher than the DoA values of all aligned concepts. The intuition is that if a replicated sentence that is composed of only one concept has a less activation value than the top-activated sentences, the unit is not sensitive to the concept compared to a sequence of different words. Using this definition of non-interpretable units, we report the layer-wise ratios of interpretable units in Fig. 9 and some examples of non-interpretable units in Fig.10 in Appendix D. Please refer to Appendix D for the detailed results.
>
>
> 4. Figure 5
> ===================================
> We thank Reviewer 2 for correcting the typo. The y-axis of Fig. 5 is “the number of aligned concepts” in each layer. For each layer, we collect all concepts, and then count category of each concept. For example, the plot on the top left dbpedia shows that more than 100 morpheme concepts are aligned to the units of the 0-th layer. We also update the caption of Fig. 5 for clarification.
>
>
> 5. Concept clusters
> ===================================
> (1) What concept clusters emerge?
> As Reviewer 2 suggested, we add experiments of concept clusters to Fig. 11 and Appendix E.1. The top and left dendrograms of Fig. 11 show the hierarchical cluster of concepts based on the vector space distance between the concepts in the last layer. For clustering ([4]), we use the Euclidean distance as the distance measure, and pretrained Glove ([1]), fastText ([2]), ConceptNet ([3]) embedding for projecting concepts into the vector space. Each element of the heat map represents the number of times two concepts are aligned in the same unit. We observe that several diagonal blocks (clusters) appear more strongly in classification than in translation, particularly in the AG News and the DBpedia dataset. Please refer to Appendix E.1 for more details.
>
> (2) Why certain clusters emerge more than others?
> It is an interesting question why certain concepts or clusters emerge more than others. We add some results to this inquiry to Section 4.5 and Appendix F. We deal with individual concepts rather than clusters of concepts. We investigate the following two hypotheses: (i) The concepts with higher frequency in training data are aligned to more units. (ii) Concepts that have more influence on the objective function (expectation of the loss) are aligned to more units. For the concepts in the final layer, we measure the Pearson correlation coefficient between [# of aligned units per concept] and the factor (i) and (ii), and obtain 0.482 / 0.531, respectively. These results make a lot of sense in that the learned representation focuses more on identifying both frequent concepts and important concepts for solving the target task.
>
> 6. Typos
> ===================================
> We corrected the typos. Thanks for pointing out.
>
> References
> ===================================
> [1] Jeffrey Pennington et al., GloVe: Global Vectors for Word Representation (EMNLP 2014)
> [2] Piotr Bojanowski et al., Enriching Word Vectors with Subword Information (TACL 2017)
> [3] Speer Robert et al., ConceptNet 5.5: An Open Multilingual Graph of General Knowledge (AAAI. 2017)
> [4] Daniel Mullner. Modern hierarchical, agglomerative clustering algorithms. arXiv:1109.2378v1. (arXiv 2011)

---

> > ### Comment · AnonReviewer2 · 2018-12-06
> > **Follow-up to author response**
> >
> > Thank you to the authors for your comprehensive replies and revisions. The added analyses help to clarify and solidify the overall picture, and I remain of the opinion that this paper offers some interesting insights into the internal workings of these networks.

---

### Meta-Review · Area_Chair1 · 2018-12-14

**Confidence:** 4
**Recommendation:** Accept (Poster)

**Metareview:**

Important problem (making NN more transparent); reasonable approach for identifying which linguistic concepts different neurons are sensitive to; rigorous experiments. Paper was reviewed by three experts. Initially there were some concerns but after the author response and reviewer discussion, all three unanimously recommend acceptance.